# Flèche versus Lunge as the Optimal Footwork Technique in Fencing

**DOI:** 10.3390/ijerph16132315

**Published:** 2019-06-30

**Authors:** Zbigniew Borysiuk, Natalia Markowska, Mariusz Konieczny, Krzysztof Kręcisz, Monika Błaszczyszyn, Pantelis T. Nikolaidis, Beat Knechtle, Paweł Pakosz

**Affiliations:** 1Faculty of Physical Education and Physiotherapy, Opole University of Technology, 45-758 Opole, Poland; 2School of Health and Caring Sciences, University of West Attica, 12243 Athens, Greece; 3Institute of Primary Care, University of Zurich, 8091 Zurich, Switzerland

**Keywords:** fencing fleche, fencing lunge, sensorimotor responses, EMG, ground reaction forces

## Abstract

The objective of the study reported in this paper involved identifying the fencing attack (flèche versus lunge) that provides greater effectiveness in a real competition. Two hypotheses are presented in the study. The first hypothesis involves the greater effectiveness of the flèche with regard to bioelectric muscular tension, and the second hypothesis involves the reduction of movement time of the flèche. Therefore, analyses were conducted by the application of EMG (electromyography) signal, ground reaction forces, and parameters representing sensorimotor responses (RT—reaction time and MT—movement time). This study included six world-leading female épée fencers (mean age: 24.6 ± 6.2 years). Throughout the procedure, the subjects performed flèche and lunge touches at the command of the coach based on visual stimuli. The experimental results indicated the greater effectiveness of the flèche compared with the lunge with regard to increases in EMG values (*p* = 0.027) in the lateral and medial gastrocnemius muscles and decreases in the duration of the movement phase (*p* = 0.049) and vertical force of the rear leg (*p* = 0.028). In conclusion, higher levels of EMG and ground reaction forces were generated during the flèche compared with the lunge, which promotes an improvement in the explosive force and contributes to a reduction in the movement phase of the entire offensive action.

## 1. Introduction

Coach observations and statistical analysis derived from high-rank tournaments demonstrate a new tendency that emerges in épée technique. Increasingly, fencers involved in a bout decide to employ the flèche instead of the lunge as the basic technique of fencing footwork. To date, fencing tactics focused on the adequate performance of a lunge, which is considered a safer solution. For example, in the latter, a riposte can be avoided if the attack is parried [1]. In the case of a failure during an attack, the fencer who initiated it has minimal chance of an effective defense in the event of an opponent’s counterattack. In the first Olympic Games in Athens in 1896, we had to deal with more static forms of footwork in fencing, such as steps, leaps towards the opponents, and the lunge itself. It was not until the 1930s that the flèche was introduced by Hungarian fencers and formed an innovative, dynamic solution in the attacks performed on a fencing strip leading to a touch [2]. For this reason, research in the field of biomechanics and motor control often applied the lunge as a diagnostic procedure in the area of comprehensive analysis [3,4,5]. Both touch techniques are fundamentally different in terms of the technique that is followed. The common point is associated with the commencement of the attack with the dominant arm holding the weapon by activating the muscles of the arm and forearm [6].

With regard to the lunge, the rear leg is applied to initiate the attack with the forward lower limb in a movement performed in the upward and forward direction. It is postulated that the attack should be performed at the instant corresponding to the maximum extension of the elbow joint and before the front foot has hit the ground [7]. In the case of the flèche, the movement pattern is initiated by the dominant arm, and the attack is executed as a result of the activity of both legs simultaneously. Of note, the forward leg is applied to initiate the pattern, and a higher load is exerted on this leg to generate significantly greater values of power. The flèche is executed in a pattern in which the fencer the touches the ground with their rear leg immediately before the attack. At this instant, this leg acts as the support as it prevents the fencer from falling [8]. Visualizations of both the lunge and flèche patterns are presented in Figure 1 and Figure 2. These movements are performed by the World Champion, Ms. Danuta Dmowska, who gave consent for the publication of her image.

We emphasize that visual and even video assessments are not capable of demonstrating the complexity of the technical structure of the touch pattern, which can only be verified by the EMG record involving the activity of the important muscles involved in these attack varieties [9]. The authors of the present work were guided by the idea that the research should be performed in conditions that are similar to the actual fencing training. On the basis of that finding, practice of the flèche and lunge was performed with the involvement of the coach in the training. The practice was designed so that the body of the coach along with the blade had the coordinates of the hit area marked with the OptiTrack system of markers. A similar procedure was applied with regard to the female fencers. In addition, individual technical tests were performed at the command of the coach to follow the procedure that is observed in a typical fencing lesson with the participation of a trainer. Since the objective of the research involved the applicatory aspect of the results, six world-leading épée fencers, who are members of the Polish national team (and four of whom won the bronze medal in the team competition at the World Championships in Leipzig in 2017), were selected to participate in the study. The innovative research system comprised OptiTrack with high-speed motion capture cameras, a 16-channel EMG, and two force plates for ground reaction force registrations that were applied to test the reactions of the rear and forward legs of the fencers. As a consequence of the synchronization of the research tools listed above, it was possible to register EMG signals, parameters of the reaction time (RT) and movement time (MT), and forces (N), simultaneously, including the possibility of graphical representation of the movement sequences.

A general hypothesis was adopted regarding the greater effectiveness of the flèche compared with the lunge. This conclusion could result from the decrease in the sensorimotor response time understood generally or due to the reduction of the duration of only one of the elements, i.e., either RT or MT. In addition, an assumption was made that higher values of the EMG signal of the examined muscles during the flèche could be attributable to the dynamic characteristics of the execution of this pattern confirmed by the parameters of the ground reaction forces. The second hypothesis regarding the positive impact of the technical structure expressed by the pattern of muscle activation on more effective coordination of the muscles during the flèche than during the lunge was confirmed by the study results.

## 2. Materials and Methods

### 2.1. Ethical Approval

The study was approved by the Bioethics Committee of the Chamber of Physicians (Resolution No. 237 of 13 December 2016), so the study was performed in accordance with the guidelines defined in the Helsinki Declaration for the conduct of clinical trials in humans.

### 2.2. Participants

The study involved six female épée fencers, who are members of national teams at advanced stages of their career (age: 24.6 ± 6.2 years). Only healthy subjects with a medical certificate including a statement regarding the possibility of safe participation in sport competition and, in particular, in training and sport competitions were selected for the study. The subjects were informed about the purpose of the research and the manner in which it is implemented. The subjects were also informed that they could withdraw from the study at any time and without giving any reasons.

### 2.3. Research Tools

The registrations of the movement patterns performed by the subjects and the coach involved a motion capture system (OptiTrack, NaturalPoint, Inc., Corvallis, OR, USA) comprising eight cameras (with a resolution of 832 × 832 px, 100 FPS) and markers employed to determine the coordinates of the body in space. On the upper body parts (i.e., the left and right sides of the subjects’ arms), the markers were located in accordance with the Plug-In Gait model [9]: Regio frontalis, regio parietalis, vertebra cervicale C7, acromion, humerus lateral epicondyle, and radius styloid. The markers were attached to the lower limbs of the subjects (on the left and right sides) in accordance with the CAST technique (calibrated anatomical systems technique) [10]: Trochanter major, spina iliaca anterior superior (ASIS), spina iliaca posterior superior (PSISI), epicondylus lateralis femoris, epicondylus medialis femoris, malleolus lateralis, malleolus medialis, calcaneus, and caput ossis metatarsi I and V. The markers were also attached to fencers’ blade and handle and to the outfit of the coach (the use of four markers determines the touch area on the trunk with the 10 cm side of each marker in addition to the makers located on the blades of the subjects and the coach).

Measurements of the bioelectric signal employed a 16-channel EMG system manufactured by Noraxon (DTS, Desktop Direct Transmission System, Noraxon, Scottsdale, ZA, USA) with a sampling accuracy of 16 bits at a frequency of 1500 Hz. The analysis involved the following muscles: Biceps brachii (BB), triceps brachii (LT), flexor carpi ulnaris (FCU), extensor carpi radialis (ECR) on the upper limb holding the weapon, biceps femoris (BF), rectus femoris (RF) on the forward leg, and gastrocnemius as well as caput laterale and caput mediale (GL, GM) on the rear leg [3,11]. The muscle activation assessment applied Ag/AgCl electrodes attached to the subjects’ bodies in accordance with the SENIAM (surface EMG for non-invasive assessment of muscles) procedures. The assessment of the ground reaction forces employed two combined Kistler force plates (Kistler type 9286AA, Kistler, Winterthur, Switzerland) operating with the sampling frequency of 1500 Hz. The synchronization of the equipment was achieved by application of TTL (Transistor–transistor logic) signals.

### 2.4. Procedure

After a warm-up, the subjects performed six attempts of a straight touch towards the trainer’s trunk. The fencers repeated the flèche (Figure 1) and the lunge patterns thrice each (Figure 2). The practice was performed with the involvement of the coach. The coach’s outfit had OptiTrack markers placed on the target field. In addition, individual technical patterns were executed at the pace given by the coach, as this procedure is standard for a typical fencing practice with the participation of the coach.

### 2.5. Data Analysis

The following variables were analyzed in the study: Sensorimotor response time for reaction time (RT), movement time (MT), bioelectric activity of the muscles, and vertical and horizontal components of the peak ground reaction forces (PGRF). The initiation of the movement pattern performed by the coach was applied as the starting signal—t_0_ (beginning of step), and the pattern was considered completed at the instant when the touch was completed—t_K_ (i.e., after a successful touch of the end of the fencing weapon on the coach’s trunk). The initial time t_0_ during the attack with a lunge performed in the direction of the marker located on the trainer’s outfit was subsequently analyzed by employing Mokka software [7]. The initiation of the movement pattern (relative to the Y axis—forward) corresponded to of the displacement of the marker on the fencer’s bodies and was considered as the initial time, t_INI_, of the movement. The reaction of the vertical component of the ground forces was interpreted to occur at the instant when the value of the signal exceeds SD ± 2.5 in relation to the mean values in the time interval from t_0_ to 100 ms.

### 2.6. Statistical Analysis

The collected data were subsequently subjected to statistical analysis by application of STATISTICA 12 software (http://statistica.www.software112.com/). Due to the lack of normality of the distributions and homogeneity of the variances of the analyzed variables, nonparametric analysis tools were applied in the present analysis. To determine the level of the significance of the differences, the non-parametric Wilcoxon test was used to determine the dependencies between the samples. In addition, Spearman’s correlation analysis was used to determine the correlation between the EMG parameters and ground force reactions.

## 3. Results

As shown in Figure 3 and Figure 4, the results derived from EMG signal analysis of the activation of particular muscles reveal that some differences were established between the results of the lunge and the flèche. Analysis of the sequence of the neuro-muscular activation of the upper and lower limbs indicates that activation of key muscles of the arms precedes activation of the muscles of the forward and rear legs. The principal difference was established for the case of the flèche with regard to the activation of the forward leg. Regarding the lunge, significant differences were established in terms of the activation of the rear leg and demonstrated by the noticeable excitation of the gastrocnemius lateralis and gastrocnemius medialis muscles.

Analysis of the details of the ground reaction forces demonstrated that during the attack executed by a flèche, the attack is initiated from the rear leg (FZ1), and the forward leg is temporarily lifted above the ground (FZ2). Subsequently, after the maximum value of the vertical and horizontal forces of the leg is obtained, this value decreases until the rear leg is completely detached from the ground. In addition, the value of the ground reaction forces of the rear leg increases (Figure 3). The vertical component of the ground reaction force in the forward leg assumes its maximum after the end of this pattern, which indicates that the rear leg leaves the ground after the hit was executed successfully. The above analysis conforms to the description of the pattern during a lunge applied throughout professional footwork technique training.

During an attack performed using a lunge, the pattern starts by a thrust from the rear lower limb (the values of the vertical components of the ground reaction forces increase), whereas the forward leg leaves the ground (Figure 4). Analysis of the data demonstrated the occurrence of statistically significant differences in terms of the movement times during the attacks executed in the patterns employing a lunge and flèche. The movement time was reduced for the attack performed by the flèche compared with that performed by the lunge (Table 1). The results also indicate statistically significant differences in the mean signal of the belly muscles of the calf. The values of the signals registered in the in the gastrocnemius lateralis and lateralis of the rear leg were greater during the attack performed with a flèche (Table 2). With regard to the ground reaction forces, the research focused was on the rear leg as it plays a key role both in the lunge and the flèche, and statistically significant differences were found in this respect in the study (Table 3).

The correlation coefficient of the value of time until the peak EMG signal and vertical and horizontal components of the peak force in the lower limbs indicates the approximately complete correlation of the vertical and horizontal components of the peak force of the rear leg and rectus femoris; the vertical component of the force registered in the forward leg, triceps brachii, and gastrocnemius lateralis and medialis; the horizontal component of the signal registered in the lower rear limb; and the activation of extensor carpi radialis longus muscle. A very high level of correlation was obtained between the time until the peak EMG signal of the extensor carpi radialis longus and the vertical component of the signal of the rear lower limb, the flexor carpi ulnaris, and the vertical component of the lower forward leg and gastrocnemius lateralis muscle and the horizontal component of the signal of the rear leg (Table 4 and Table 5).

During the touch completed with a lunge, approximately complete correlation of the mean time values until the peak EMG signal and vertical and horizontal components of the forces was established between the extensor carpi radialis longus muscle and the horizontal component of the signal of the forward leg. The correlation was very high between the following: Biceps brachii and the vertical component of the signal in the rear leg, extensor carpi radialis longus, biceps femoris, triceps surae, and horizontal component of the force signal recorded in the rear leg. Due to the total detachment of forward lower limb from the ground during the lunge, the correlation between the mean time values until the peak with the vertical component of the signal registered in the lower limb was not taken into consideration.

## 4. Discussion

The research results demonstrate that the temporal aspects need to involve the difference between the components of the sensorimotor responses with regard to the lunge and flèche [12,13], which particularly applies to movement time (*p* = 0.046). The mean values recorded in the six investigated fencers with regard to MT were as follows: 510 ms (flèche) and 568 ms (lunge). By contrast, RT was almost identical, i.e., 175 ms for the flèche and 173 ms for the lunge. This fact indicates that information processing (identification, selection, programing of reaction) in response to visual stimuli requires a similar amount of time, whereas the duration of the movement for the flèche is reduced by as much as 58 ms [9,11]. This remark indicates an important tactical advantage results from the decision to execute a flèche given that fencing speed bias in épée fencing is equal to 80 ms. The fencer needs to precede the opponent’s move such that the electronic referee apparatus records the touch signaling with the ignition of a single light [14,15].

In addition, by application of the EMG, the analysis of the sequence of neuro-muscular activation of key muscles was performed by application of the example of a single selected fencer. The structure of the muscle tension sequences in relation to the arm and forearm holding the weapon is similar both during the flèche and lunge. In accordance with the input provided by other studies [16,17], the pattern is initiated by the triceps muscle in the arm acting as the extensor, followed by the activation of extensors and flexors of the carpi radialis muscle. This motion is followed by activation of the biceps femoris for the flèche, whereas the characteristic phenomenon of the lunge is associated with a clearly delayed response of the biceps femoris, which is preceded by stimulation of the gastrocnemius lateralis and gastrocnemius medialis muscles in the rear leg.

The standard routine followed during a fencing attack involves the initiation of the attack with the arm holding the weapon, and this research indicates that throughout the sequence of activation of the lower limbs in the flèche, it is more important to note the activity of the rear leg, whereas we need to pay attention to the activity of the forward leg in a lunge. The differences found in this study could have a significant practical value in the sense of identifying a successful fencing action in specific conditions. During the fencing bout, the ability to observe preliminary signals originating from the lower limbs may contribute to accurate anticipation of the type of the attack followed by the application of a defense technique as a result of adopting an adequate position and distance [18,19].

In this respect, interesting input is provided by the analysis of the EMG signal. We note that by comparing the values of the bioelectric activity of the examined muscles in the flèche and the lunge, with the exception of the triceps lateralis, the parameters demonstrate higher values for the EMG signal in the activity involving a flèche. This observation is statistically significant with respect to the gastrocnemius lateralis and the medialis muscles of the rear leg. This fact does not contradict the above-stressed conclusion about the significance, namely, the advantage of the forward leg in the attack executed with a flèche. We importantly conclude that during the course of the fleche, the rear limb is activated in advance, and the gastrocnemius lateralis and gastrocnemius medialis in the limb provide the entire effect of the momentum and the needed power during the attack [20,21]. This conclusion is complemented by the results of the ground reaction forces in the rear and forward legs recorded during the flèche.

The results confirm that the attack executed from the rear lower limb is sequentially combined with the detachment of this foot from the ground. Then, after the maximum values of the vertical and horizontal forces of the lower rear limb are obtained, the value decreases until the foot is completely lifted off the ground. At this moment, the touch is accomplished, and the values of the ground reaction forces of the lower limb increase. This action acts as a support to prevent the attacking fencer from falling, which is forbidden in accordance with the regulations of the fencing competition. Thus, the final activation of the forward limb ensures that the balance is maintained after the entire technical pattern executed by the flèche. With regard to the lunge, such a spectacular sequence of the muscular stimulation of the limb was not recorded with regard to the correlation of forward and rear legs with the ground reaction forces. When a touch is executed in a pattern using the lunge, the lift-off effect is obtained by employing muscles in the rear leg, which is confirmed by the increase of the vertical ground reaction forces and is further confirmed by the detachment of the foot from the ground.

The analysis of the correlations between EMG parameters and ground reaction forces demonstrates significant dependences between them in both forward and rear legs. We note high levels of correlation during the flèche (0.96) between the activation of the rectus femoris of the forward leg and the gastrocnemius lateralis (0.94) of the rear leg. For the lunge, correlation levels of (0.82) and (0.85) were established with regard to the gastrocnemius lateralis and the medialis of the rear leg, respectively. These results demonstrate a significant advantage of the flèche over the lunge in terms of dynamic characteristics and forces generated by the muscles involved in the process [22]. In addition, it is reasonable to note that, in the conditions of significant correlations between EMG and ground reaction forces, it is possible to apply both test procedures alternatively without facing the loss of reliability of the resulting data.

## 5. Conclusions

On the basis of the assumptions presented in the introduction section of this paper, the following conclusions can be formulated. The increased effectiveness of the flèche compared with the lunge in contemporary fencing results from the time constraints involving the decrease of the duration of the MT (movement time). The RT (reaction times) are similar to the times recorded in other footwork patterns. During the flèche, higher values of EMG and components of the ground reaction forces were generated by the investigated fencers compared with the lunge, which led to an increase in the explosiveness and reduced the duration of movement phase of the attack. This innovative research tool is capable of synchronizing the operation of force plates to register ground reactions with EMG and OptiTrack and demonstrates that the flèche is executed at a similar distance as the lunge. In addition, the touch is executed at the instant corresponding to the increased activation of the rear limb before the final stimulation of the forward leg. This fact provides new light on the elements of coordination training during the flèche.

## Figures and Tables

**Figure 1 ijerph-16-02315-f001:**
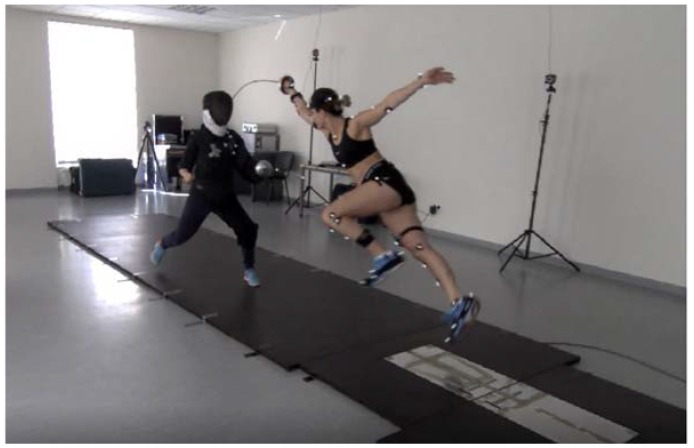
Straight touch employing flèche.

**Figure 2 ijerph-16-02315-f002:**
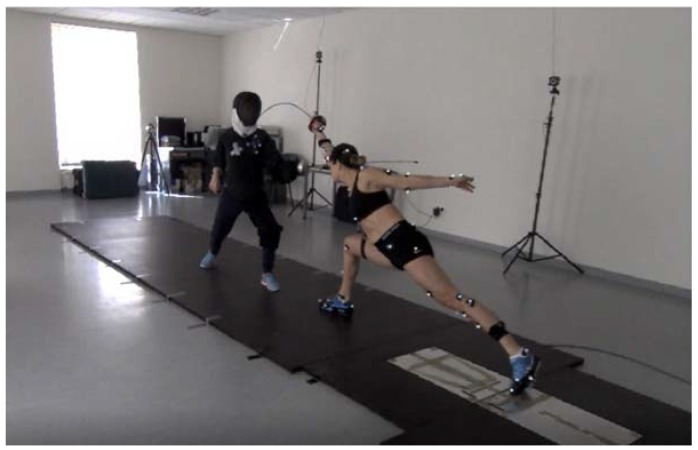
Straight touch employing lunge.

**Figure 3 ijerph-16-02315-f003:**
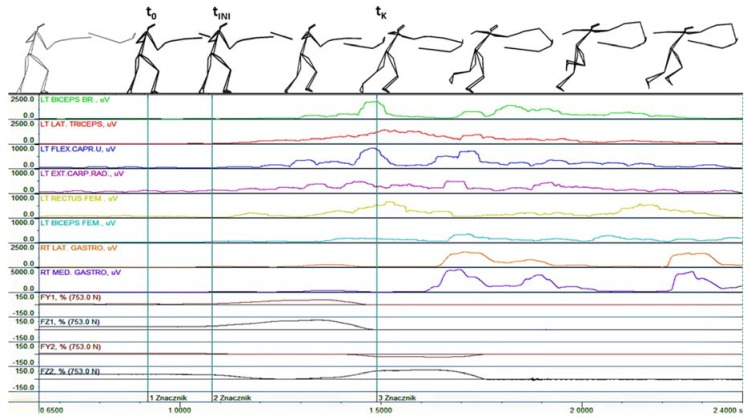
Bioelectric activity of selected muscles as well as the horizontal (FY1 and FY2) and vertical components (FZ1 and FZ2) of the ground reaction forces during a touch completed by a flèche in response to visual stimulation; 1—t_0_: Initiation of the move by the coach, 2—t_INI_: Fencer’s response, 3—t_K_: Instant corresponding to completion of the flèche. FY: horizontal component of ground reaction forces, FZ: vertical component of ground reaction forces.

**Figure 4 ijerph-16-02315-f004:**
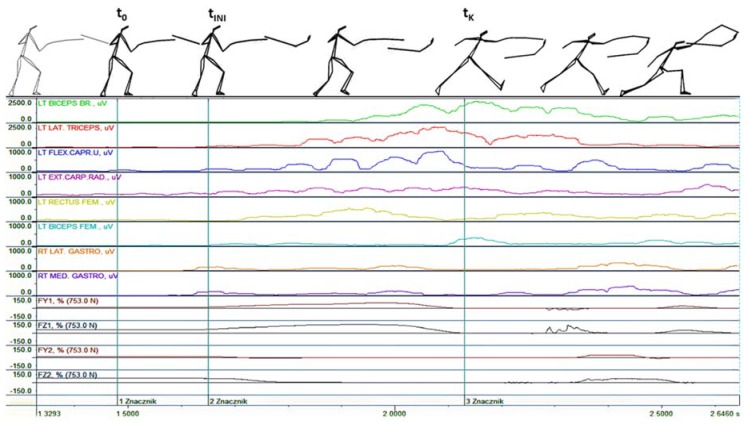
Bioelectric activity of selected muscles as well as the horizontal (FY1 and FY2) and vertical components (FZ1 and FZ2) of the ground reaction forces during an attack executed with a lunge in response to visual stimulation; 1—t_0_: Initiation of the move by the coach, 2—t_INI_: Fencer’s response, 3—t_K_: Instant of completion of the lunge.

**Table 1 ijerph-16-02315-t001:** Comparison of the mean values of the standard deviation of the reaction and movement times during an attack executed by the flèche and the lunge.

Times	Flèche		Lunge		
	Mean	Stand. dev.	Mean	Stand. dev.	*p*-Values
RT (ms)	0.175	0.025	0.173	0.018	0.916
MT (ms)	0.51	0.036	0.568	0.039	0.046 *

RT—reaction time, MT—movement time, * *p* ≤ 0.05, stand.: standard, dev.: deviation.

**Table 2 ijerph-16-02315-t002:** Comparison of the mean values of the standard deviation of muscular tension during attacks executed by the flèche and the lunge.

Muscular Tensions	Flèche		Lunge		*p*-Values
	Mean	Stand. dev.	Mean	Stand. dev.	
Biceps brachii (µV)	252.6	107.1	226.2	114.3	0.6
Lat. triceps (µV)	192	84.5	213.9	170.3	0.6
Flex. carp. u (µV)	194.6	58.5	176	56.6	0.345
Ext. carp. rad. (µV)	192.3	39	178.8	43	0.345
Biceps femoris (µV)	73.1	25.6	51.3	21.6	0.074
Lat. gastro. (µV)	146 *	33.1	73.1 *	37.9	0.027
Med. gastro. (µV)	207.7 *	79	94.7 *	59.7	0.027
Rectus femoris (µV)	151.1	18.4	139.3	44.2	0.463

Lat.: lateral, Flex.: flexor, carp.: carpi, Ext.: extensor, carp.: carpi, rad.: radialis, Lat.: lateral, gastro.: gastrocnemius, Med.: medial, gastro.: gastrocnemius, * *p* ≤ 0.05.

**Table 3 ijerph-16-02315-t003:** Comparison of the mean values of the standard deviation of the ground reaction forces during attacks executed by the flèche and the lunge.

Ground Reaction Forces	Flèche		Lunge		Flèche Visual Stimulus vs. Lunge Visual Stimulus
	Mean	Stand. dev.	Mean	Stand. dev.	*p*-Value
FZ_rear_ (*N*)	142.8	23.6	115.9	13.3	0.028

FZ_rear_: vertical component of ground reaction forces.

**Table 4 ijerph-16-02315-t004:** Correlation coefficients of the mean values until the peak electromyography (EMG) signal for the vertical (FZ) and horizontal components (FY) of the signal recorded in the rear leg and forward legs during attacks executed by the flèche in the interval from t_0_ to t_K_.

Flèche	FZ_rear leg_	FZ_forward leg_	FY_rear leg_
	Correlation Coefficient		
Biceps brachii (ms)	−0.42	−0.27	−0.45
Lat. triceps (ms)	0.54	0.92 *	0.57
Flex. carp. u (ms)	0.32	0.81 *	0.39
Ext. carp. rad. (ms)	0.87 *	0.81	0.92 *
Biceps femoris (ms)	0.75	0.46	0.81
Lat. gastro. (ms)	0.79	0.94 *	0.85 *
Med. gastro. (ms)	0.79	0.76	0.81
Rectus femoris (ms)	0.94 *	0.78	0.96 *

* *p* ≤ 0.05.

**Table 5 ijerph-16-02315-t005:** Correlation coefficients of the mean values of the times until the peak EMG signal was recorded for the vertical (FZ) and horizontal components (FY) of the signal recorded during a touch executed with a lunge.

Lunge	FZ_rear leg_	FY_rear leg_
	Correlation Coefficient	
Biceps brachii (ms)	0.19	0.85
Lat. triceps (ms)	0.89 *	0.79
Flex. carp. u (ms)	0.64	0.71
Ext. carp. rad. (ms)	0.53	0.98 *
Biceps femoris (ms)	0.49	0.88 *
Lat. gastro. (ms)	0.67	0.85 *
Med. gastro. (ms)	0.65	0.82 *
Rectus femoris (ms)	−0.03	0.16

* *p* ≤ 0.05.

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
