# Peer review of "Flèche versus Lunge as the Optimal Footwork Technique in Fencing"

_ijerph, 2019, doi:10.3390/ijerph16132315_

Round 1
Reviewer 1 Report
Dear authors
the paper is interesting, only some minor aspects should be revised:
- in abstract please define the abbreviation before to use it (EMG, MT, RT, MG, lat. and med.)
- from line 64 - 80 should be moved and adapted in method section
- the background of the study it seems to be not exhaustive, more rational and analysis of the literature is necessary to formulate the objectives of the study
- the reviewer suggest to join table 2 with table 4
Author Response
Comments and Suggestions for Authors
Dear authors
The paper is interesting, only some minor aspects should be revised:
- In abstract please define the abbreviation before to use it (EMG, MT, RT, MG, lat. and med.)
Answer: Generally, we would like to emphasize the fact that the following abbreviation mainly “MG” has been corrected as the “EMG”. Additionally, the lack of expressions of interpretation for the two further abbreviations which were used such as “RT” and “MT” which have been developed as the “reaction time” and “movement time
- From line 64 - 80 should be moved and adapted in method section
Answer: I consider no need for the text lines 64-80 to be moved to the method section. In my opinion the fragment mentioned above presents one of the main ideas of the work to perform and measure technical actions with the active participation of the coach in the experiment. That causes research more practical and closer to the real duels in fencing. My point justifies keeping the explanation discussed into introduction part.
- The background of the study it seems to be not exhaustive, more rational and analysis of the literature is necessary to formulate the objectives of the study
Answer: I partly agree to this opinion, however, this kind of research should be still considered as pioneering and they have no reflection in the contemporary literature of its kind.
- The reviewer suggest to join table 2 with table 4
Answer: From the technical perspective we definitely could not join the tables numbered 2 and 4 due to the fact that they concern completely different parameters.
Reviewer 2 Report
This particular manuscript sought to evaluate biomechanical differences in two different fencing attacks. The authors reported that the fleche is more explosive, and has a lower movement phase. This manuscript is really well written, but a few alterations could make it even better.
Abstract: Line 19. RT and MT are not defined at this point in time. Line 23, neither is MG.
Abstract: Line 22. I would suggest that 'proved' is a little strong.
Page 4, line 135. RT and MT were defined on page 3, line 79, so this is not necessary.
Page 4, Section 2.6 How was normality assessed? What software was used to evaluate the data?
Author Response
Comments and Suggestions for Authors
This particular manuscript sought to evaluate biomechanical differences in two different fencing attacks. The authors reported that the fleche is more explosive, and has a lower movement phase. This manuscript is really well written, but a few alterations could make it even better.
Abstract: Line 19. RT and MT are not defined at this point in time. Line 23, neither is MG.
Answer: The mistake indicated as “MG” has been corrected in the paper as “EMG”. Further abbreviations such as “RT” and “MT” have been defined as “Reaction time” and “movement time”.
Abstract: Line 22. I would suggest that 'proved' is a little strong.
Answer: The term “proved” has been softened and replaced by the “indicated” term.
Page 4, line 135. RT and MT were defined on page 3, line 79, so this is not necessary.
Answer: We suggest maintaining the mentioned abbreviations in both fragments the abstract and the main text as well.
Page 4, Section 2.6 How was normality assessed? What software was used to evaluate the data?
Answer: The “normality” has been defined applying the Shapiro-Wilk’s test. The software used to evaluate the data was Statistica 12.
Reviewer 3 Report
This is very nicely written and high impact manuscript.
Very minor edits are.
Line 35: Please fix reference format
Line 56: Please fix the sentence/grammar.
All the best.
Author Response
Comments and Suggestions for Authors
This is very nicely written and high impact manuscript.
Very minor edits are.
Line 35: Please fix reference format
Answer: Thanks, reference format has been fixed.
Line 56: Please fix the sentence/grammar.
Answer: Thanks, sentence has been fixed.